# Life Experiences with Using Community Care among People with Severe Physical Disabilities: A Comparative Analysis between South Korea and Japan

**DOI:** 10.3390/ijerph17249195

**Published:** 2020-12-09

**Authors:** Minyoung Lee, Yeji Choi, Eun Young Lee, Dong-A Kim, Seung Hee Ho

**Affiliations:** 1Department of Healthcare and Public Health Research, National Rehabilitation Research Institute, Seoul 01022, Korea; interlaw88@korea.kr (M.L.); chldp7@korea.kr (Y.C.); eun9246@korea.kr (E.Y.L.); 2Division of Public Medical Rehabilitation, National Rehabilitation Center, Seoul 01022, Korea; kana69@nrc.go.kr

**Keywords:** community care, pathway mapping, interpretive phenomenological analysis, care planning, independent living, people with severe disability

## Abstract

This study identified the pathways chosen by people with severe physical disabilities (PWSPD) in South Korea and Japan in using community care throughout their life and compared their experiences while navigating these pathways from their perspective. A concurrent nested mixed-method design was adopted. Quantitative data analysis included pathway mapping of facilities and services used throughout their lives. For qualitative data, interpretative phenomenological analysis (IPA) was applied. Eleven South Korean (congenital 7, acquired 4) and nine Japanese (congenital 6, acquired 3) participants were surveyed and interviewed. Pathway mapping was conducted by classifying the participants into focus groups. South Korean participants took nine years more than the Japanese participants to reach independence and showed different pathway characteristics. Superordinate themes from the IPA provided insight into the differences in experiences between PWSPD of the two countries: (1) accessibility and continuity of medical services; (2) experience of vocational training; (3) way and degree of social support for independent living; (4) care planning for receiving comprehensive services. In developing a community care model for the PWSPD to accelerate their time to independence, the government should strive for accessibility and connectivity of medical services, strengthen vocational training, social support for independent living, and information provision for the PWSPD.

## 1. Introduction

In response to a projected increase in the aging population and the resulting increase in medical expenses, many countries have redesigned their services to support the elderly to remain at home for a longer duration [1,2]. This “redesigning” entails a shift in resources from acute care to community care to provide improved support to enhance the ability to live in one’s own home and community safely, independently, and comfortably regardless of age, income, or ability level [3].

Since South Korea entered the aged society in 2017 and is projected to enter an ultra-aged society in 2025, their government is working to develop a community care model not only for the elderly but also for the people with disabilities (PWD) [4]. However, the model appears to focus mainly on the elderly and does not adequately consider the situation and needs specific to the PWD.

In South Korea, PWD have difficulty returning to the local community after being discharged; therefore, they often travel to several hospitals. Subsequently, their medical costs were estimated to be about three times higher as compared to the general population [5]. If they become elderly without being independent through appropriate rehabilitation training and social support, the social burden will increase manifold.

Particularly, as observed in several cases during the coronavirus disease (COVID-19) pandemic, people with severe physical disabilities (PWSPD) living in long-term residential care facilities or isolated in their house experienced increased social isolation, low access to medical and healthcare services, were unable to receive prompt rescue, and were avoided by activity assistants in an emergency situation. This situation is contrary to the Convention on the Rights of Persons with Disabilities, an international human rights treaty of the United Nations intended to protect the rights and dignity of persons with disabilities. In particular, the Convention devotes article 19 to the right to live independently and be included in the community. Therefore, PWSPD should be the main target population for community care.

In designing the community care model for PWSPD, a patient-centered approach emphasizing their experiences would provide a deeper understanding of their needs during the transition to independence as well as the factors that affect their independent living in the community. Several previous studies have dealt with the experiences of PWD in hospitals and communities [6,7,8,9,10,11,12]. However, most studies have reported localized and segmented experiences in specific situations, such as receiving rehabilitation [6,7], attempting to return to work [8,9,10], or transitioning from pediatric to adult healthcare [11,12]; therefore, obtaining an integrated idea for designing a community care model is difficult.

This study aimed to (1) comprehensively identify the pathways taken by of PWSPD in using community care, such as medical, health, and welfare services in related facilities, throughout their life, and (2) explore their experiences while walking through these pathways from their perspective. Particularly, we compared the pathways and experiences of PWSPD in South Korea with those in Japan that had introduced and implemented community care 10 years earlier than South Korea and has similar medical and welfare legal systems, to find concrete implications for designing the community care model for PWSPD.

## 2. Materials and Methods

### 2.1. Study Design

A concurrent nested mixed-method design, an approach when doing mixed methods research that requires data to be collected in parallel within the same study and qualitative method dominates while the quantitative method is embedded or “nested” within, was adopted [13]. It is increasingly acknowledged that a triangulated approach merging both quantitative and qualitative data shows more multifaceted results than analyzing either data type separately [14]. For quantitative data, descriptive statistics and pathway mapping of the use of facilities and services throughout their life journey was conducted. For qualitative data, interpretative phenomenological analysis (IPA) [15,16] was applied to understand the (1) key components of the experience of PWD in using the facilities and services (e.g., needs, challenges, barriers, motivations) and (2) differences in the experiences between Korean and Japanese participants. IPA has its theoretical origins in phenomenology, hermeneutics, and idiography. In this study, the phenomenological component referred to the lived experiences of PWSPD, and the idiographic component referred to the use of facilities and services in the targeted areas in Korea and Japan. Hermeneutics involves understanding the identified differences in the experiences between Korean and Japanese participants through the analysis and interpretation of interview data [17].

### 2.2. Participants

We used purposive sampling to recruit PWSPD in the target regions of South Korea and Japan. Hence, we partnered with the centers for independent living (CIL) in the targeted regions of each country. We requested the National Rehabilitation Center for Persons with Disabilities (NRCD) in Japan to provide recommendations for target regions. At least 10 participants from each country were deemed appropriate to address both the quantitative and qualitative components of the present study [18].

The inclusion criteria were: (1) having a severe disability as defined by the law of each country (i.e., levels 1–2 out of 6 levels given in the Welfare of Disabled Persons Act of South Korea [19]; level 5–6 out of 6 levels given in the Services and Supports for Persons with Disabilities Act of Japan [20], (2) aged below 65 years, (3) working for the CIL, and (4) having sufficient cognitive ability to participate in a focus group interview. We recruited 22 PWSPD (congenital 7 and acquired 5 in South Korea; congenital 7 and acquired 3 in Japan) and assessed them for eligibility according to the inclusion criteria; however in each country, one individual was dropped for not meeting the inclusion criteria, leaving 20 enrolled participants.

Participants were classified into the following subgroups considering the social and environmental differences in life that they would have experienced from the onset of their disability: (1) South Korean participants with congenital disability (Group KC), (2) South Korean participants with acquired disability (Group KA), (3) Japanese participants with congenital disability (Group JC), and (4) Japanese participants with acquired disability (Group JA). The National Rehabilitation Center of South Korea (NRC) approved this study (NRC-2019-03-015). Written informed consent was obtained from all participants, following the Declaration of Helsinki.

### 2.3. Data Collection

Basic demographic and disability-related data were collected through a survey conducted before the interview (Appendix A). Data included the type of utilized facilities and services, the time taken to begin using them, and the period of use, focusing on building a timeline of the participants’ experiences from the onset of disability to the current point. Facilities were classified into four different types: medical, health, welfare, and education facilities. Additionally, the number of personal assistant supported a month by the government were asked. The law of each country regulates the number of personal assistant for PWPSD and their supporting time of the physical, housework and social activity for PWPSD depending to the severity of the disability. Interviews were scheduled according to the participant’s convenience in each country. All the interviews were conducted by one author, Korean and Japanese bilingual, with over 5 years of qualitative research experience in the field of rehabilitation and healthcare. The individual interviews were conducted for about 60–90 min at first, followed by 3–4–additional written or telephone interviews when necessary. Semi-structured interviews were developed to explore the facilities and services they utilized at each stage of the life journey framework, and how they experienced and felt in utilizing them. Questions were also posed to elicit responses about their perceived enablers, difficulties, or challenges encountered in seeking independence throughout their life journey.

In this study, we adopted the five stages of life journey framework, developed by Sezgin and colleagues based on key models from the stages of change literature [21] and the transtheoretical model [22]: (1) diagnosis (the time period when people learn that their perceived symptoms are related to a disability), (2) initial adjustment (the time period when people learn to create structures to manage therapy routines and their lives), (3) early transition (the time period when people begin the process of transitioning to independence), (4) mid-transition (the time period when people build long-term perspectives to better understand and manage their condition and life in the long run), (5) late transition (the final stage in transitioning to independence). However, we modified the framework to replace the initial adjustment stage with the school-age stage in the case of individuals with congenital disability, and regarded the late transition stage as the time when people started to work for the CIL.

### 2.4. Data Analysis

For quantitative analysis, the type of facilities utilized at each stage, the time taken to begin using them, and the period of use were individually described. The pathways in using the facilities and services were individually mapped and presented by group. The average time taken to enter the late transition stage from diagnosis was calculated by group.

For the qualitative analysis, IPA was used to explore how participants comprehended their personal and social circumstances through their lived experiences in using the facilities and services [15], and what differences existed between the experiences of participants of the two countries. Each individual transcript of the interview was analyzed in three steps according to the guidelines suggested by Smith and colleagues [15]. In the first stage, the researcher conducted descriptive coding, identifying key descriptions and emotional responses. The second step involved phenomenological coding, focusing on how and what part of the textual data contributed to understanding the content and meaning behind the participant’s words. In the third stage, interpretative coding was undertaken to identify patterns that would allow the researcher to develop themes in the next stage of analysis. Thereafter, the researcher searched for emerging themes across all cases within each group. Two more researchers reviewed and analyzed the interview transcripts using the same process to improve the consistency of the analytical procedures. Discrepancies were resolved through discussion among researchers. After the analysis, member checking [23] was undertaken to improve credibility and trustworthiness [17].

## 3. Results

We recruited 22 PWSPD (congenital 7 and acquired 5 in South Korea; congenital 7 and acquired 3 in Japan) and assessed them for eligibility according to the inclusion criteria; however in each country, one individual was dropped for not meeting the inclusion criteria, leaving 20 enrolled participants. Participants with congenital disability had diseases, such as cerebral palsy, spinal muscular atrophy, muscular dystrophy, spinal dysplasia, and pediatric brain infarction, while those with acquired disability had spinal cord injury. Table 1 presents demographic and disability-related information. Appendix B provides additional individual information about utilized facilities, the time taken to begin using the facility and the period of use.

### 3.1. Findings-Pathway Mapping

Journey maps were developed to demonstrate the pathways taken by the participants of each group (Figure 1, Figure 2, Figure 3 and Figure 4). The key findings are summarized below by group.

#### 3.1.1. Group JC: Japanese Participants with Congenital Disability

(1)**Diagnosis-School-age stage:** Four out of six participants (JC2, JC3, JC5, and JC6) had received continuous comprehensive medical and care services from infancy to adulthood, not only in the rehabilitation hospital, but also in facilities such as the National Rehabilitation Center for Children and Disabilities (NRCCD), a facility for children with physical disabilities, and a nursery school. JC4 received medical services in the rehabilitation hospital for 2 years from the age of 6, when he had a pediatric brain infarction. After that, he utilized home visiting services until now. JC1 was not aware that she had muscular atrophy until 25 years of age, and therefore, did not receive any rehabilitation training until then. However, she had visited hospitals to identify her disease.(2)**Early transition stage:** Two participants (JC1 and JC2) who graduated from a regular high school went to a university, whereas other participants who graduated from a special school worked at a sheltered workshop for 9 years (JC4) or received vocational rehabilitation at the National Vocational Rehabilitation Center for the Persons with Disabilities (NVRCD) for 1 year (JC5 and JC6). From this stage, participants began to establish a care plan and received corresponding services, except JC1, who did not notice her disability until 25 years of age.(3)**Middle transition stage:** JC1 worked from home while JC2, JC3, and JC4 participated in a program for independent living in supported housing organized by the CIL. This stage was omitted for JC5 and JC6, because they had started working for the CIL at the end of the early transition stage.(4)**Late transition stage:** Participants’ average duration from the onset of disability to the entry point to the late transition stage was 26 years (range, 19–40 years). Four participants lived alone (JC1, JC2, JC3, and JC4) and utilized visiting medical, nursing, or rehabilitation services as well as 24-h personal assistance service (PAS). JC6 began using housework aid after marriage.

#### 3.1.2. Group JA: Japanese Participants with Acquired Disability

(1)**Diagnosis-Initial adjustment stage:** Participants were admitted to the acute care hospital to receive acute medical treatment for 6–12 months.(2)**Early transition stage:** Immediately after discharge from the acute care hospital, all three participants were admitted to the NRCD for 6–12 months, involving rehabilitation training to improve the activities of daily living. During the NRCD and after discharge, JA3 had worked from home as a furniture designer for 2 years. Participants began to establish a care plan and received corresponding services from this stage.(3)**Middle transition stage:** JA1 and JA2 received additional training of daily living specific for PWSPD in the National Rehabilitation Center for Persons with Severe Disabilities (NRCSD) for 2 to 3 years. Afterwards, JA1 utilized the day-care center for 3 years, while JA2 worked from home for 2 years. Both participated in the 3-day supported housing program organized by the CIL. For JA3, this stage was omitted because she had begun working for the CIL at the end of the early transition stage.(4)**Late transition stage:** Participants’ average duration from the onset of disability to the entry point to the late transition stage was 8.17 years (range, 3–16 years). At this stage, all the participants utilized visiting medical, nursing, or rehabilitation services and 24-hour PAS.

#### 3.1.3. Group KC: South Korean Participants with Congenital Disability

(1)**Diagnosis-School-age stage:** Group KC was classified into three categories: a person (1) who attended a regular school (KC1), (2) who attended a special school for the PWD (KC2, KC3, KC4, and KC5), and (3) who did not attend any school (KC6 and KC7). Some participants among those who attended a special school received medical services in the university hospital to which the special school belonged. However, people who did not attend any school could not receive any medical services until adulthood.(2)**Early transition stage:** All the participants who attended a regular and special school went to a university, except KC5, who had worked in a sheltered workshop for 20 years since middle school. Two uneducated participants (KC6 and KC7) had stayed at home without using any facilities or services until they entered their 20s: KC6 moved to a supported housing at the age of 29 enrolling in the independent living program and started to utilize the rehabilitation hospital simultaneously, while KC7 stepped into social life after receiving vocational rehabilitation in a sheltered workshop at the age of 20.(3)**Middle transition stage:** After graduating from the university and leaving the sheltered workshop, participants chose various pathways, such as (1) working from home (KC1), (2) going to a graduate school (KC2), (3) going to a private institute to learn employment skills (KC3), and (4) participating in the independent living program in a supported housing (KC4, KC5, and KC6). Two uneducated participants (KC6 and KC7) began taking courses in the online university after taking the general equivalency diploma over 30 years of age.(4)**Late transition stage:** Participants’ average duration from the onset of disability to the entry point to the late transition stage was 35.14 years (range, 27–47 years). Most of the participants utilized the PAS, except for KC5. Four participants utilized the welfare center for the PWD for participating in a recreation program or receiving a hairdressing service.

#### 3.1.4. Group KA: South Korean Participants with Acquired Disability

(1)**Diagnosis-Initial adjustment stage:** Participants were admitted to the acute care hospital for 1 to 1.5 years.(2)**Early transition stage:** About 5 years after discharge from the acute care hospital, KA1 and KA2 were shortly admitted to the NRC for 2–3 months to receive activities of daily living and driving rehabilitation. KA3 was admitted to the university hospital for 6 months to receive bladder control training. KA4 returned to graduate school, which he had attended before having the disability, and began to work for the CIL shortly after graduation.(3)**Middle transition stage:** After discharge from the NRC and the university hospital, KA1 and KA3 had stayed at home for over 15 years and did not use any facilities and services, other than admission to the rehabilitation hospital for 3 months (KA1) and going to the college (KA3). KA2 had been in long-term residential care facilities for 15 years, and then moved to a supported housing for independent living, while simultaneously using visiting nursing services and PAS.(4)**Late transition stage:** Participants’ average duration from the onset of disability to the entry point to the late transition stage was 17.5 years (range, 5–20 years). At this stage, all the participants utilized the PAS.

### 3.2. Findings-Interpretative Phenomenological Analysis

Superordinate themes from the IPA are presented to provide insight into the differences in the experiences by group: (1) accessibility and continuity of medical services, (2) the experience of vocational training, (3) the way and degree of social support for independent living, and (4) care-planning for receiving comprehensive services.

#### 3.2.1. Accessibility and Continuity of Medical Services

Most participants of Group JC regularly received comprehensive medical and care services not only in a local rehabilitation hospital but also in a nursery school and facility for children with physical disabilities from infancy to adulthood. They felt a sense of satisfaction in that they could receive continuous rehabilitation services and that the medical and welfare services were connected at these institutions.


*“I have been attending the NRCCD since I was 2 years old. My family doctor was there, and in that case, I could keep using the facility. The center linked the in-hospital orthopaedic surgeon to the wheelchair company and helped me manufacture a wheelchair. They also provided information on scoliosis and respiratory diseases as well as prescribed medication and rehabilitation services once a month.”*
*(Participant JC2)* 

However, one participant who had been receiving rehabilitation services at a facility for children with physical disabilities complained of dissatisfaction with the old-fashioned uniform rehabilitation training and insufficient training amount.


*“I have been using facility for children with physical disabilities from the age of 4 to the present. It is a place that provides rehabilitation services as well as nursing and medical care on behalf of the family. However, since the only treatment was to forcefully stretch the contracted muscles in the old way, I only remember the pain and it seemed to have no effect. Even that, I could not get enough because of the lack of staff.”*
*(Participant JC3)* 

Among the participants of Group JA who received medical services at the NRCD immediately after discharge from the acute hospital, some participants were satisfied with the rehabilitation service focused on returning to the community, whereas others felt that rehabilitation training suitable for PWSPD who had low levels of motor function was insufficient.


*“There were many people, such as doctors and therapists, who knew my disability well at the NRCD, so I was able to learn the knowledge of appropriate countermeasures, and received rehabilitation services aimed at returning to the community such as how to use a wheelchair.”*
*(Participant JA1)* 


*“At the NRCD, PWSPD who had low levels of motor function, like me, could not get enough training for activities of daily living, so there was not much I could do on my own.”*
*(Participant JA2)* 

After starting their social life, most Japanese participants received visiting services. They experienced immense satisfaction in that the visiting services had flexible timings and provided the appropriate service at home, although they felt uncomfortable about not being able to use those visiting services during the weekends.


*“Having someone waiting for me 24 hours a day is not only very physically convenient, it also gives me an indescribable sense of security. It is very helpful to have a nurse who is familiar with my disability and give me medical advice that my helper cannot. The only regret is that the visiting service is conducted only during the daytime on weekdays, so it is difficult to use for those who work.”*
*(Participant JA2)* 


*“A visiting doctor comes twice a month on a regular basis, but depending on the contract, they come anytime when I get sick. A doctor taking care of me now had a terminal medical care contract with me, so he is supposed to come and look after me whenever I call him until I die.”*
*(Participant JA3)* 

In Group KC, some participants who attended a special school located in a rehabilitation hospital received rehabilitation treatment while attending the special school. However, after graduating from the special school, most of the participants could not receive any rehabilitation treatment because of institutional constraints, absence of personal assistants, and inaccessibility to medical services.


*“When I was in special school, I received physical and occupational therapy at the hospital inside the school, but after that, I could not receive any treatment. Now my function declines as I get older, so I want to receive visiting rehabilitation service to maintain function... But I am not receiving it because I am not a recipient of basic living (which serves as a condition for receiving visiting rehabilitation service from a public community center).”*
*(Participant KC2)* 


*“I have not used medical services since special school because I needed to do physical therapy myself using equipment in the treatment room (not receiving therapy by the therapists). But I could not do well because I was always lying down. It was difficult to ask my mother to help me because she was too old.”*
*(Participant KC3)* 

One participant in Group KC had no experience in receiving rehabilitation treatment at all from infancy to adulthood because there was no facility available in the vicinity and family members could not provide help to move around.


*“I have not received physical or occupational therapy until I entered the supported housing for independent living at the age of 29 because there was no place to receive service around… and my parents were busy doing business.”*
*(Participant KC6)* 

Participants in Group KA received a short 6-month rehabilitation 3 to 10 years after the acute stage or could not receive rehabilitation at all due to lack of availability of facilities in the vicinity and customized services, restrictions on mobility, and burden of medical expenses.


*“I needed more (physical therapy after discharge), but I could not get it… If I had been near to the rehabilitation hospital, I would have been able to comfortably get physical therapy, or if the road to there was easy to move using a wheelchair, I would have used it (physical therapy) at a certain cost. But I felt It’s just annoying because it’s not accessible.”*
*(Participant KA3)* 


*“I think the most necessary thing for my health is physical therapy. But even if I go to the hospital, there were not many wheelchair fitness equipment customized for me. I luckily found one rehabilitation hospital that had the fitness equipment imported directly from the Netherlands that suited me. But, soon, the hospital was closed because it was not economically maintained.”*
*(Participant KA1)* 

One participant who was admitted to a long-term residential care facility for 15 years used the nursing service, but was unable to perform rehabilitation training because there was no manpower to provide rehabilitation or equipment suitable for her in the facility.


*“There was a rehabilitation equipment in a long-term residential care facility, but I did not use it because it did not fit my body, kept hurting my body.”*
*(Participant KA2)* 

#### 3.2.2. The Experience of Vocational Training

For Group JC, those who graduated from a regular high school entered college, while those who graduated from a special high school received vocational training at the NVRCD or a sheltered workshop. The latter were generally satisfied, and received a job at the CIL right after leaving the NVRCD.


*“(After graduating from special school), I moved to Saitama where the NVRCD is located to receive vocational training. The NVRCD is different from a hospital in that PWD can stay for up to five years and concentrate on vocational training. I learned computer skills and how to calculate money. Actually, I am not currently doing anything using such skills, but it helped me to learn the concept of money.”*
*(Participant JC6)* 

Meanwhile, one participant who had worked at a sheltered workshop for nine years after graduating from a special school complained about not being able to communicate with the non-disabled on an equal footing.


*“There was a perception that it was natural for those who graduated from special school to go to a sheltered workshop, so there was no other way to find a career that suited my ability. At a sheltered workshop, I was often treated as a child without being able to communicate on an equal footing with non-disabled people.”*
*(Participant JC4)* 

In Group JA, 2 out of 3 participants were admitted to the NRCSD for 2–3 years after being discharged from the NRCD. The participants were satisfied with their life in the NRCSD in that they could receive specialized rehabilitation training focused on severe disabilities and meet colleagues with similar levels of motor function.


*“The NRCSD is a specialized facility for training for independent living of PWSPD, and 99% of residents are patients with cervical spinal cord injury. The NRCSD has specialized know-how such as the way of customized nursing care and housing remodelling suited to one’s physical function, so it is a place where PWD and their family can naturally accept disabilities. Many doctors often recommend that we (patients with cervical spinal cord injury) go there after discharge.”*
*(Participant JA3)* 


*“Since the NRCSD is an admission facility for people with the same type of disability, it was naturally possible to share disability-related information and imagine one’s situation after discharge, seeing a person who had been discharged earlier, which motivated us to participate in rehabilitation training.”*
*(Participant JA2)* 

Participants were able to work at the CIL very quickly after the training period; however, it was not easy to work at institutions other than the CIL due to severe disability and institutional restrictions.


*“I even completed a master’s program to go to a general company related to IT service, but if I got a job, I should give up care services supported by the government, such as visiting care services for people with severe disabilities. Since these systems are designed to support the daily lives of PWD, it is forbidden to use helpers supported by this system while studying at school or earning income at a company. Further, a full-time job was physically too tight for me.”*
*(participant JC2)* 

In Group KC, irrespective of graduating from a regular or special high school, participants continued to invest in additional education, such as going to graduate school or private academy after graduating from college. However, despite the extended study, it was difficult to find a job.


*“I tried my best after graduating from an online university (to find a job), but I could not get a job. I looked for a vocational training academy, but the academy also refused because of severe disability.”*
*(Participant KC6)* 


*“My major was advertising, and I could not get a job right after graduating from graduate school because I could not keep myself steady enough to work. So, after graduation, I paid my tuition to learn video editing (at a private academy). I think I did it for about 8 years, but I could not find a job related to the advertising.”*
*(Participant KC3)* 

There were rare cases where they succeeded in finding a job soon after graduating from the university by receiving employment information from the CIL.


*“When I lived in the supported housing for independent living, I was introduced the CIL from a head hunter company that provided employment support. The job coach of the company guided me.”*
*(Participant KC4)* 

Two participants, one graduated from a special school and the other was uneducated, worked in a sheltered workshop but they did not consider it as a vocational training.


*“I went to a special school until middle school, but the special high school was opened a little later, so I received job training in a sheltered workshop (instead of entering high school). (Omitted) Actually, I just folded my shopping bag rather than receiving vocational training.”*
*(Participant KC5)* 

In Group KA, no one attended institutions for vocational training after discharge from the acute care hospital. However, similar to Group KC participants, they also made additional investments in education, such as going to a university. The participants themselves thought that vocational training was impossible for PWSPD.


*“I thought that vocational training was hardly applicable to people with cervical spinal cord injury, so I studied this and that by myself, and after taking the general equivalency diploma, I entered the university majored in social welfare.”*
*(Participant KA3)* 


*“I did not go to the institutions for vocational training. No way. I cannot even move my fingers how can I do vocational training?”*
*(Participant KA1)* 

#### 3.2.3. The Way and Degree of Social Support for Independent Living

To support the independent living of the PWD, PAS and supported housing systems existed in both countries. However, the quantitative degree and focus of the service content were somewhat different.

All Japanese participants were receiving the 24-hour PAS, except for two participants living with their families, and they felt a sense of security and freedom by using the service. One participant stated that the reason he could live independently despite having a severe disability was the 24-hour PAS.


*“Thanks to the 24-hour PAS, I can do what I want to do when I want to. It creates a sense of security because I believe that whatever happens, my helper would take care of it, and I also feel secure because one helper continuously provides services without member change. Also, it is good that the helper fits my lifestyle.”*
*(Participant JA2)* 

In Korea, the establishment of the PAS system was relatively late, and in the past, there were restrictions on going wherever they wanted in the community. After the system was established, restrictions on mobility were reduced; however, there were some participants who felt limited in their activities due to mobility problems.


*“Somehow, I could not go to the welfare center or gym because I could not be free to move.”*
*(Participant KA4)* 

Some Japanese participants began living alone through CIL’s 3-month supported service program for independent living and a 3-day short experience of living in a supported house.


*“Suddenly, I left home and started living alone. Two months before living alone, I was trained to live alone in the CIL-supported house for 3 days and 2 nights.”*
*(Participant JC2)* 

By comparison, South Korean participants lived in the supported house for a relatively long period of 1.5–7 years, although they felt that the quality of the service program in the supported house was not high. However, nonetheless, they found it more satisfactory than living in the facilities or with their parents.


*“Training for independent living in the supported house was training in words, but actually just living together. Rather than experiencing independent living, I just felt protected. However, nonetheless, I felt freer than living with my parents... I was satisfied just because I could do it myself.”*
*(Participant KC6)* 

#### 3.2.4. Care Planning for Receiving Comprehensive Services

Group JC participants received consultation support by the care manager at the counseling support center for the PWD for making a career decision after graduating from school. Often, there were cases in which participants recognized new necessary services while receiving consultation for care planning or receiving services based on the care plan.


*“The care manager at the counseling support center wrote my care plan, so I was able to live in a dormitory at NVRCD. For the application of living aid services, the care manager also assisted me in writing the service plan.”*
*(Participant JC6)* 


*“In order to apply the job search support service, I was supported by a care manager of a counseling support center in writing a service plan. In the process of finding a job through the job search support service, I became aware of the existence of the CIL and made me want to live independently.”*
*(Participant JC5)* 

In Group JA, some participants received care planning support from the counseling support center when they were discharged from the hospital to find facilities and services that suited their circumstances, while other participants established self-care planning by themselves.


*“Currently, I conducted a self-care plan, but before I became independent from home, I did not know how to (care planning), so I set up a plan for support for mobility and physical care through the counseling support center. It was nice that the care manager came to my house and consulted with me to write a service plan and read it to me after writing.”*
*(Participant JA2)* 

Some South Korean participants lacked timely information and were unable to receive education or disability-related services at the time of need.


*“When I was young, I am sorry that I did not know that special schools provide specialized education for the PWS. I even did not know the fact that there are such educational institutions.”*
*(Participant KC7)* 


*“Seven years after the disability occurred, I could not live with my family at my house, so I went to the facility. When I went to the facility, I learned that there was a disability level for the first time, so I got a disability level. If I had received the disability level early, I would have been able to get various benefits sooner... I did not know that.”*
*(Participant KA2)* 

One participant stated that the lives of people with congenital disability, especially in childhood, depended on their parents’ will and knowledge.


*“The general perception of people is that it is natural for people with disabilities to go to special schools, and only in some cases where parents were informed that people with disability could also attend regular schools could go to regular schools. At the time, it was important what the parents thought and from whom the parents could get the advice.”*
*(Participant KC1)* 

## 4. Discussion

We identified the pathways chosen by PWSPD of South Korea and Japan in using facilities and services throughout their life journey and compared their experiences while walking through these pathways from their perspective. Specifically, this study is meaningful in that it derives implications for designing a community care model for the PWSPD.

Through pathway mapping, we found a remarkable difference between the two countries: South Korean participants’ duration from the diagnosis of disability to independence was about nine years longer than that of the Japanese participants. The characteristics of the pathways of Japanese participants, which were considered to have contributed to this difference, include: (1) a care plan established at the point of early transition enabled them to discover and use the facilities and services appropriate to their function and situation, (2) intermediate facilities specialized for the PWSPD utilized at the time of early transition helped them return to the community, and consequently, the middle transition period could almost be omitted or ended shortly, and (3) visiting services and 24-hour PAS supported them to start their social life earlier and maintain it stably.

According to the qualitative analysis, four superordinate themes provided insight into the differences in experiences between participants of the two countries. First, there was a difference in the accessibility and continuity of medical services. Japanese participants with congenital disabilities received comprehensive medical and care services from infancy to adulthood. Noticeably, the linkage of medical and welfare services and activation of visiting services are seen as key factors that contributed to the accessibility and continuity of medical services for the PWSPD. In South Korea, most participants rarely accessed medical services, except those who intermittently used the NRC or rehabilitation hospital for a short period of less than 6 months after the initial adjustment and school-age period. According to the previous studies, poor continuity of care, disengagement from healthcare, and lack of integration between pediatric and adult healthcare systems have been reported as the PWD and caregivers’ perceived barriers to successful healthcare [11,24]. South Korean participants recognized that this inaccessibility and discontinuity was due to institutional constraints, lack of availability of facilities and customized services, restrictions on mobility, and burden of medical expenses.

Recently, the South Korean government has made efforts to create an environment in which the PWD can easily access medical services in the community by newly establishing a general physician system specific for the PWD and a medical expenses support program. It is necessary to seek a way to link medical and welfare services, such as connecting welfare services at hospitals or allowing medical services to be received at welfare facilities, considering the Japanese participants and results of previous studies [11,24]. Additionally, the expansion of target audience for visiting services and establishment of regulations allowing medical fees specific for visiting services needs to be considered because until now visiting services have been applied only to the recipient of the national basic livelihood security program and lower income class through the public community center.

Second, concerning the experience of vocational training, Japanese participants could identify their capabilities in social life through vocational training in a specialized public institution. It was also naturally possible to share disability-related information and imagine their situation following discharge after witnessing a previously discharged person, which motivated them to participate in rehabilitation training. The Services and Supports for Persons with Disabilities Act of Japan defines vocational training for the PWD, and 34,356 facilities delivered vocational training for increasing the knowledge and competence necessary for employment in 2018 [25]. In contrast, we often found that South Korean participants continued an additional investment in the private sector of education, such as going to private academies or graduate schools because they could not find suitable vocational training facilities for the PWSPD. The Act on the Employment Promotion and Vocational Rehabilitation of Persons with Disabilities of South Korea also defines services similar to those defined by Japan. However, the problem lies in the lack of wide implementation of these services. There were only 15 facilities nationwide for vocational training in 2018 [26]. In the National Survey in 2017, 94.5% of the respondents reported that they had no vocational training experience [27], disproving the lack of implementation despite the regulation for vocational training by the law. Therefore, reinforced legal and administrative measures should be taken so that the regulated services can be practically operated. Particularly, there is a need to establish more facilities that provide vocational training specialized for the PWSPD, not only to provide specialized rehabilitation for independence but also to promote information exchange between peers. This is consistent with the results of previous studies reporting that peer support helps PWD to alleviate fears around returning to work, provide motivation to be a worker, and assist with designing employment opportunities [8,10,28,29]

Third, concerning social support for independent living, both countries operated the PAS, the difference being in the time of application and the method of manpower management. According to the results of the current study, Japanese participants who lived alone were receiving assistance from 11 to 18 personal assistants through the 24-hour PAS. They felt satisfied with the system in that they could do what they wanted when they wanted, expressing that the system was the reason they could be independent despite severe disabilities. In contrast, South Korean participants were receiving assistance from 1 to 4 personal assistants for an average of 399.43 h a month, often saying that they were restricted from their activities even after the PAS was established. The reason for this difference might be the different legal regulations for the type of personal service in the two countries: the law of South Korea regulates that an activity assistant should be provided to PWD who have difficulty in daily and social life [19]; however, unlike Japan, mobility assistance is not specified separately, and therefore it can be difficult for PWD to ask for help from their personal assistants when going out within limited support hours. Mobility restrictions could create barriers to access other services or social participation, as seen in the experiences of PWD reported in other studies [24,30]. Therefore, for the independence of PWD in both daily and social life, mobility assistance besides activity assistance in PAS should be separately specified and personal assistance time and manpower should be increased.

Lastly, there was a difference in the navigation of disability-related information between participants of the two countries, depending on the presence or absence of a care planning system. Japanese participants received counseling support to establish a comprehensive care plan from the counseling support center or conducted self-care planning mostly at the end of the early transition period, and received services based on the established care plan. This is because the 2012 amendment in the law stated that the governments should deliver disability-related services only when a care plan is prepared [20]. We can presume that the early intervention of the government through a care planning support system would accelerate the time to independence of the PWSPD. In contrast, in South Korea, the use of services was often dependent on their parents’ capability of information acquisition up to school age, and differences in information obtained during this period had a great influence on the participants’ pathways. The time to enter the early transition was generally quite delayed, since they could not acquire enough and appropriate information about the resources available to them in the community, and since they struggle alone, the pathways during the mid-transition period were distributed very diversely. In previous studies, participants with cerebral palsy and their caregivers regarded the opportunities lost in transition and roadmap to care as problems in the transition from pediatric to adult services and requested a transparent transition plan [12]. In another study regarding experiences with navigating information in the community, some participants with SCI mentioned that they missed health improvement opportunities due to a lack of information provision and experienced frustration at the uncertainty of their future [31]. These results highlight the need for developing a systematic care planning support system including specialized facilities and manpower.

This study has several limitations. First, there would be differences in systems related to the disability, including facilities and services, depending on the region, even within each country. Therefore, caution should be maintained while interpreting the results of the current study as a general situation of South Korea and Japan. Second, although we attempted to match the participants of each country in the demographic and disability-related characteristics, there may still be some differences in socio-economic aspects. Moreover, since we compared the pathways and experiences of participants throughout their lives, it should be considered that the facilities and services that South Korean participants have not used in the past could exist today. Four, we adopted a concurrent nested mixed-method design, which qualitative method dominated while the quantitative method was embedded to the small size of participants, so the external validity of the findings and the generalizability of the study to other people is compromised.

## 5. Conclusions

PWSPD in South Korea took more time from the diagnosis of disability to independence as compared to Japanese participants. Experiences of PWSPD of the two countries differed in terms of accessibility and continuity of medical services, the experience of vocational training, the way and degree of social support for independent living, and care planning for receiving comprehensive services. The government deserves to consider supplementing the facilities and services that are identified as insufficient in developing a community care model for PWSPD in order to accelerate the time to independent living of the PWSPD. To be specific, we recommend the linkage of medical and welfare services, the activation of visiting services, and the establishment of vocational training centers specialized for the PWSPD, 24-hour PAS and a care planning system.

## Figures and Tables

**Figure 1 ijerph-17-09195-f001:**
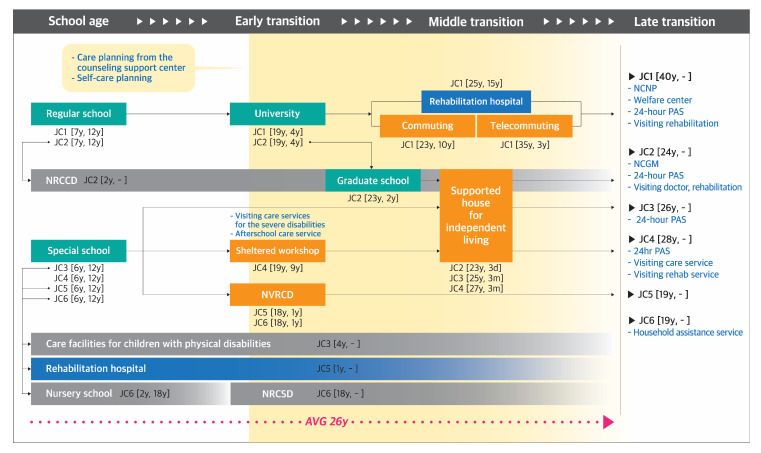
Pathway mapping for Japanese individuals with congenital disability. NRCCD, National Rehabilitation Center for Children and Disabilities; NVRCD, National Vocational Rehabilitation Center for Persons with Disabilities; NRCSD, National Rehabilitation Center for Persons with Severe Disabilities; NCNP, National Center of Neurology and Psychiatry; NCGM, National Center for Global health and Medicine); PAS, personal assistance services. **Note.** Blue, orange, orange, green and grey color in the square respectively indicate facilities providing medical, welfare, education, and mixed medical and welfare related services. Code and number indicate participant code [the time taken to begin using the facility, the period of use] (Ex. JA2 [7y, 12y]).

**Figure 2 ijerph-17-09195-f002:**
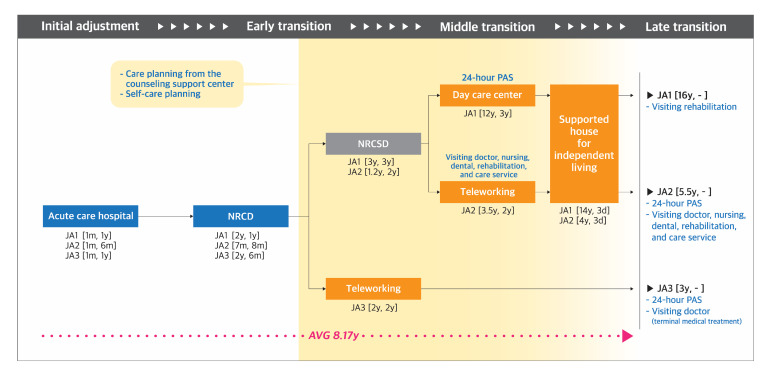
Pathway mapping for Japanese individuals with acquired disability. NRCD, National Rehabilitation Center for Persons with Disabilities; NRCSD, National Rehabilitation Center for Persons with Severe Disabilities; PAS, personal assistance services. **Note.** Blue, orange, orange, green and grey color in the square respectively indicate facilities providing medical, welfare, education, and mixed medical and welfare related services. Code and number indicate participant code [the time taken to begin using the facility, the period of use] (Ex. JA1 [3y, 3y]).

**Figure 3 ijerph-17-09195-f003:**
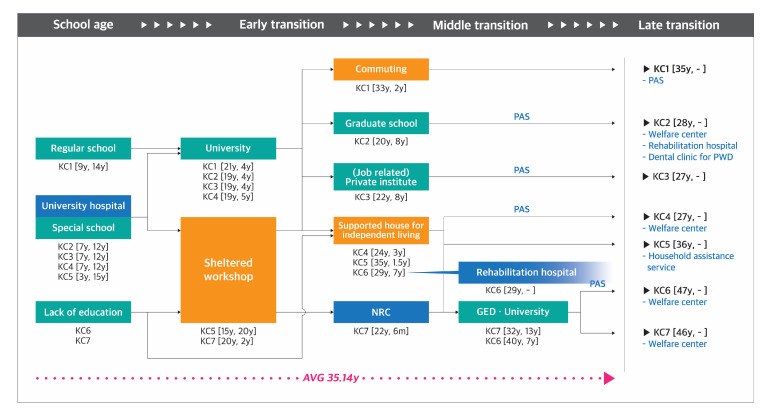
Pathway mapping for South Korean individuals with congenital disability. NRC, National Rehabilitation Center; GED, general equivalency diploma; PAS, personal assistance services. **Note.** Blue, orange, orange, green and grey color in the square respectively indicate facilities providing medical, welfare, education, and mixed medical and welfare related services. Code and number indicate participant code [the time taken to begin using the facility, the period of use] (Ex. KC7 [22y, 6m]).

**Figure 4 ijerph-17-09195-f004:**
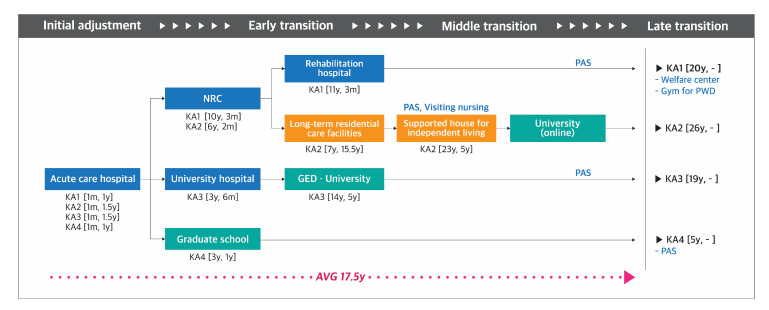
Pathway mapping for South Korean individuals with acquired disability. NRC, National Rehabilitation Center; GED, general equivalency diploma; PAS, personal assistance services. **Note.** Blue, orange, orange, green and grey color in the square respectively indicate facilities providing medical, welfare, education, and mixed medical and welfare related services. Code and number indicate participant code [the time taken to begin using the facility, the period of use] (Ex. KA1 [11y, 3m]).

**Table 1 ijerph-17-09195-t001:** Demographic and disability-related information of the participant.

Participant Code	Acquired/ Congenital Disability	Title of Disease	Gender	Age	Cohabitant	Num of Personal Assistant
JC1	C	Muscular dystrophy	F	64	Spouse, children	11
JC2	C	Spinal muscular atrophy	M	27	None	14
JC3	C	Cerebral palsy	F	49	None	16
JC4	C	Pediatric brain infarction	M	33	None	16
JC5	C	Spinal dysplasia	M	23	Parents	0
JC6	C	Cerebral palsy	F	22	Spouse	2
JA1	A	Spinal cord injury (C6)	F	50	None	18
JA2	A	Spinal cord injury (C5)	M	32	None	14
JA3	A	Spinal cord injury (C6)	F	51	None	15
KC1	C	Cerebral palsy	M	43	None	1
KC2	C	Cerebral palsy	F	40	None	1
KC3	C	Cerebral palsy	F	34	Parents	1
KC4	C	Cerebral palsy	M	36	None	3
KC5	C	Cerebral palsy	F	43	None	0
KC6	C	Cerebral palsy	M	57	Spouse	3
KC7	C	Cerebral palsy	F	48	None	1
KA1	A	Spinal cord injury (C6)	M	52	Spouse	1
KA2	A	Spinal cord injury (C3,4,5)	F	51	colleague	3
KA3	A	Spinal cord injury (C4,5)	M	45	None	2
KA4	A	Spinal cord injury (C6,7)	M	50	None	4

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
