# Peer review of "Life Experiences with Using Community Care among People with Severe Physical Disabilities: A Comparative Analysis between South Korea and Japan"

_ijerph, 2020, doi:10.3390/ijerph17249195_

Round 1

Reviewer 1 Report

Authors reported well written and easy to read research paper that is addressed to “comprehensively identify the pathways taken by of PWSD in using community care, such as medical, health, and welfare services in related facilities, throughout their life, and (2) explore their experiences while walking through these pathways from their perspective”.

This aim appears to be critically important for development of design of community healthcare system. In this context this article that is summarized the findings received from both South Korean and Japanese populations are attractive for readers.

The abstract contains concise description for aim, methods, results and conclusion. The methods completely match to both the aim and results. Authors found that PWSD in South Korea took more time from the diagnosis of disability to independence as compared to Japanese participants. Yet, they gave the recommendation to the governments of both countries to supplement the facilities and services that are identified as insufficient and consider them.

The conclusions are consistent with the evidence and arguments, which are presented in the article.

The strength of the study was a use of appropriate pathway mapping for several individuals with acquired disability to evaluate differences in accessibility and continuity of medical services, the experience of vocational training, the way and degree of social support for independent living, and care planning for receiving comprehensive services for both countries. The weaknesses of the study were heterogeneity of both population and substantial difference in healthcare system in both countries.

Novelty of the study is clear and concise description of the difference in medical services, the experience of vocational training, and the navigation of disability-related information between participants of the two countries, depending on the presence or absence of a care planning system.

Authors defined limitations of their study and widely disputed them. Overall, these findings are original and novel and deserve to be published.

Methodology of the study in quite high, whereas there are some items need to be explained.

Major issues: NONE

Minor issues:

  1. Authors should add scheme regarding the facilities and services that can be used by governments to improve community care model for PWSD
  2. Section Methods. Please, add information about the study population, size of population, criteria for enrollment of the data, because the section Results contains limited information about it.

Author Response

Revision Notes (ijerph-1016322)

The authors would like to express our appreciation to the editor and reviewer for their considered efforts in reviewing this paper and for all their most valuable comments. We have done our best to incorporate the editor’s and reviewers’ recommendations in the revised version of our paper.

The changes are marked in red in the revised version of the manuscript.

Responses to the Reviewer 1

Comments to the Author

Authors reported well written and easy to read research paper that is addressed to “comprehensively identify the pathways taken by of PWSD in using community care, such as medical, health, and welfare services in related facilities, throughout their life, and (2) explore their experiences while walking through these pathways from their perspective”.

This aim appears to be critically important for development of design of community healthcare system. In this context this article that is summarized the findings received from both South Korean and Japanese populations are attractive for readers.

The abstract contains concise description for aim, methods, results and conclusion. The methods completely match to both the aim and results. Authors found that PWSD in South Korea took more time from the diagnosis of disability to independence as compared to Japanese participants. Yet, they gave the recommendation to the governments of both countries to supplement the facilities and services that are identified as insufficient and consider them.

The conclusions are consistent with the evidence and arguments, which are presented in the article. The strength of the study was a use of appropriate pathway mapping for several individuals with acquired disability to evaluate differences in accessibility and continuity of medical services, the experience of vocational training, the way and degree of social support for independent living, and care planning for receiving comprehensive services for both countries. The weaknesses of the study were heterogeneity of both population and substantial difference in healthcare system in both countries.

Novelty of the study is clear and concise description of the difference in medical services, the experience of vocational training, and the navigation of disability-related information between participants of the two countries, depending on the presence or absence of a care planning system.

Authors defined limitations of their study and widely disputed them. Overall, these findings are original and novel and deserve to be published. Methodology of the study in quite high, whereas there are some items need to be explained.

Major issues: NONE

Reviewer 1’s comment 1:

Authors should add scheme regarding the facilities and services that can be used by governments to improve community care model for PWSD

Response to Reviewer’s comment 1:

Thank you for your kind suggestion. In response to your comments, we added scheme regarding the facilities and services that can be used by governments to improve community care model for PWSD as follows:

Page 16, Lines 605-607 [Conclusions]

To be specific, we recommend linkage of medical and welfare services, activation of visiting services, and establishment of vocational training centers specialized for the PWSPD, 24-hour PAS and a care planning system.

Reviewer 1’s comment 2

Please, add information about the study population, size of population, criteria for enrollment of the data, because the section Results contains limited information about it.

Response to Reviewer1’s comment 2:

Thank you for your kind comment. In response to your comments, we clarified the study population, size of population, criteria for enrollment of the data in the Materials and Methods and Results section:

Page 2-3, Lines 92-102 [Materials and Methods]

At least 10 participants from each country were deemed appropriate to address both the quantitative and qualitative components of the present study [18].

The inclusion criteria were: (1) having a severe disability as defined by the law of each country (i.e., levels 1-2 out of 6 levels given in the Welfare of Disabled Persons Act of South Korea [19]; level 5-6 out of 6 levels given in the Services and Supports for Persons with Disabilities Act of Japan [20], (2) aged below 65 years, (3) working for the CIL, and (4) having sufficient cognitive ability to participate in a focus group interview. We recruited 22 PWSPD (congenital 7 and acquired 5 in South Korea; congenital 7 and acquired 3 in Japan) and assessed them for eligibility according to the inclusion criteria; however in each country, one individual was dropped for not meeting the inclusion criteria, leaving 20 enrolled participants.

Page 4, Lines 158-161 [Results]

We recruited 22 PWSPD (congenital 7 and acquired 5 in South Korea; congenital 7 and acquired 3 in Japan) and assessed them for eligibility according to the inclusion criteria; however in each country, one individual was dropped for not meeting the inclusion criteria, leaving 20 enrolled participants.

Reviewer 2 Report

This manuscript describes about  Life Experiences with Using Community Care among People with Severe Disabilities in South Korea and Japan. Also, describes the community care, accessibility and continuity of medical services, vocational training and degree of social support for independent living in two different countries. The methodology are describes clearly and results are presented well. I have very few minor comments are below:

Page 2 and Line 145: Please correct the total number of persons chosen/screened for the study and total number included in the study after inclusion/exclusion for better clarity.

As per page 2, Line 145, Japanese Participants with Congenital Disability are only 3 and as per line 156 and table 1, JC participants are 4, please clarify.

Author Response

Responses to the Reviewer 2

Comments to the Author

This manuscript describes about Life Experiences with Using Community Care among People with Severe Disabilities in South Korea and Japan. Also, describes the community care, accessibility and continuity of medical services, vocational training and degree of social support for independent living in two different countries. The methodology are describes clearly and results are presented well. I have very few minor comments are below:

Reviewer 2’s comment 1:

Page 2 and Line 145: Please correct the total number of persons chosen/screened for the study and total number included in the study after inclusion/exclusion for better clarity.

Response to Reviewer2’s comment 1:

Thank you for your kind comments. In response to your comments, we corrected the sentences as follows:

Page 2-3, Lines 92-102 [Materials and Methods]

The inclusion criteria were: (1) having a severe disability as defined by the law of each country (i.e., levels 1-2 out of 6 levels given in the Welfare of Disabled Persons Act of South Korea [19]; level 5-6 out of 6 levels given in the Services and Supports for Persons with Disabilities Act of Japan [20], (2) aged below 65 years, (3) working for the CIL, and (4) having sufficient cognitive ability to participate in a focus group interview. We recruited 22 PWSPD (congenital 7 and acquired 5 in South Korea; congenital 7 and acquired 3 in Japan) and assessed them for eligibility according to the inclusion criteria; however in each country, one individual was dropped for not meeting the inclusion criteria, leaving 20 enrolled participants.

Page 4, Lines 158-161 [Results]

We recruited 22 PWSPD (congenital 7 and acquired 5 in South Korea; congenital 7 and acquired 3 in Japan) and assessed them for eligibility according to the inclusion criteria; however in each country, one individual was dropped for not meeting the inclusion criteria, leaving 20 enrolled participants.

Reviewer 2’s comment 2

As per page 2, Line 145, Japanese Participants with Congenital Disability are only 3 and as per line 156 and table 1, JC participants are 4, please clarify.

Response to Reviewer2’s comment 2:

As you pointed out exactly, in the previous manuscript the number of persons with congenital disabilities and the number of persons with acquired disabilities were listed in reverse. Now, we corrected it as follows:

Page 1, Lines 17-18 [Abstract]

Eleven South Korean (congenital 7, acquired 4) and nine Japanese (congenital 6, acquired 3) participants were surveyed and interviewed.

Page 2-3, Lines 92-102 [Materials and Methods]

At least 10 participants from each country were deemed appropriate to address both the quantitative and qualitative components of the present study [18].

The inclusion criteria were: (1) having a severe disability as defined by the law of each country (i.e., levels 1-2 out of 6 levels given in the Welfare of Disabled Persons Act of South Korea [19]; level 5-6 out of 6 levels given in the Services and Supports for Persons with Disabilities Act of Japan [20], (2) aged below 65 years, (3) working for the CIL, and (4) having sufficient cognitive ability to participate in a focus group interview. We recruited 22 PWSPD (congenital 7 and acquired 5 in South Korea; congenital 7 and acquired 3 in Japan) and assessed them for eligibility according to the inclusion criteria; however in each country, one individual was dropped for not meeting the inclusion criteria, leaving 20 enrolled participants.

Page 4, Lines 158-161 [Results]

We recruited 22 PWSPD (congenital 7 and acquired 5 in South Korea; congenital 7 and acquired 3 in Japan) and assessed them for eligibility according to the inclusion criteria; however in each country, one individual was dropped for not meeting the inclusion criteria, leaving 20 enrolled participants.

Reviewer 3 Report

In my opinion, this a very consistent and interesting paper comparing access by persons with disabilities to independent living in Japan and South Korea. From my point of view this paper can be published, and I have only some minor suggestions to improve it:

  1. It could be a good idea to include in the Introduction a brief reference to the UN Convention on the Rights of Person with Disabilities, which is today the main normative framework of disability policies. In fact, the Convention devotes article 19 to the right to live independently and be included in the community, which has a close relation to the topic of the paper.
  2. I think that the word "disabled", which is sometimes used in the article, should be avoided, firstly because the Convention makes a clear option for the expression "persons with disabilities", and second because the use of the word "disabled" seems to place in the forefront the disability not considering the person. So, for example, in line 292 the expression "severely disabled people" should be replaced by "people with severe disabilities", or even better, in my opinion, "persons with severe disabilities".
  3. In line 510 it can be read that "the introduction of telemedicine should be reviewed in this COVID-19 era". On the issue of telemedicine there is a wide discussion in this moment, specially from the ethical point of view. It is a complicated issue which should be analyzed more deeply and cannot be only mentioned in one sentence as if it were not problematic. This sentence does not add anything to the content of the article, and therefore I would suggest deleting it.
  4. My most important suggestion concerns the conclusions. I think they are extremely short and only contain a synopsis of the ideas that have already appeared in the discussion. In my opinion, it would be much more interesting to formulate some policy recommendations based on the discussion developed in section 4, which could be also of interest for other countries different from Japan and South Korea.

Author Response

Responses to the Reviewer 3

Comments to the Author

In my opinion, this a very consistent and interesting paper comparing access by persons with disabilities to independent living in Japan and South Korea. From my point of view this paper can be published, and I have only some minor suggestions to improve it:

Reviewer 3’s comment 1:

It could be a good idea to include in the Introduction a brief reference to the UN Convention on the Rights of Person with Disabilities, which is today the main normative framework of disability policies. In fact, the Convention devotes article 19 to the right to live independently and be included in the community, which has a close relation to the topic of the paper.

Response to Reviewer3’s comment 1:

Thank you so much for your kind suggestion. In response to your suggestion, we added a reference to the UN Convention on the Rights of Person with Disabilities in the Introduction as follows:

Page 2, Lines 51-54 [Introduction]

This situation is contrary to the Convention on the Rights of Persons with Disabilities, an international human rights treaty of the United Nations intended to protect the rights and dignity of persons with disabilities. In particular the Convention devotes article 19 to the right to live independently and be included in the community. Therefore, PWSPD should be the main target population for community care.

Reviewer 3’s comment 2

I think that the word "disabled", which is sometimes used in the article, should be avoided, firstly because the Convention makes a clear option for the expression "persons with disabilities", and second because the use of the word "disabled" seems to place in the forefront the disability not considering the person. So, for example, in line 292 the expression "severely disabled people" should be replaced by "people with severe disabilities", or even better, in my opinion, "persons with severe disabilities".

Response to Reviewer3’s comment 2:

In response to your comments, we avoided the word “disabled” and replaced that word with “people with severe physical disability (PWSPD)”, or “people with disability (PWD) as follows:

Page 9, Lines 307 [Results]

Among the participants of Group JA who received medical services at the NRCD immediately after discharge from the acute hospital, some participants were satisfied with the rehabilitation service focused on returning to the community, whereas others felt that rehabilitation training suitable for PWSPD who had low levels of motor function was insufficient.

Page 10, Lines 311 [Results]

“At the NRCD, PWSPD who had low levels of motor function, like me, could not get enough training for activities of daily living, so there was not much I could do on my own.” (Participant JA2)

Page 13, Lines 478 [Materials and Methods]

“When I was young, I am sorry that I did not know that special schools provide specialized education for the PWS. I even did not know the fact that there are such educational institutions.” (Participant KC7)

Reviewer 3’s comment 3:

In line 510 it can be read that "the introduction of telemedicine should be reviewed in this COVID-19 era". On the issue of telemedicine there is a wide discussion in this moment, specially from the ethical point of view. It is a complicated issue which should be analyzed more deeply and cannot be only mentioned in one sentence as if it were not problematic. This sentence does not add anything to the content of the article, and therefore I would suggest deleting it.

Response to Reviewer3’s comment 3:

Thank you for your kind comments. We fully agree with your opinion and deleted the sentence as follows:

Page 14, Lines 526 [Discussion]

Moreover, the introduction of telemedicine should be reviewed in this COVID-19 era.

Reviewer 3’s comment 4:

My most important suggestion concerns the conclusions. I think they are extremely short and only contain a synopsis of the ideas that have already appeared in the discussion. In my opinion, it would be much more interesting to formulate some policy recommendations based on the discussion developed in section 4, which could be also of interest for other countries different from Japan and South Korea.

Response to Reviewer3’s comment 4:

Thank you for your kind suggestions. In response your comments, we added some policy recommendations in the Conclusions based on the discussion as follows:

Page 16, Lines 604-606 [Conclusions]

The government need supplement the facilities and services that are identified as insufficient and consider them while developing a community care model for PWSPD that could accelerate the time to independent living. To be specific, we recommend linkage of medical and welfare services, activation of visiting services, and establishment of vocational training centers specialized for the PWSPD, 24-hour PAS and a care planning system.

Reviewer 4 Report

This study examines the pathways taken by people with severe disabilities in using community care in two different settings (Japan and South Korea) using a mixed-method design. The paper is interesting and provides new information. The mapping technique used to illustrate the different pathways between the different groups is visual and interesting. While the results might be indicative of something interesting, there are several additional concerns in the research design that need to be justified. My primary concern relates to the very small sample size, with heterogeneous samples (e.g., most of the South Korean sample present with cerebral palsy, while only two participants from Japan present with this condition).

  1. The authors stated that at least 10 participants from each country were needed, which may be appropriate for the qualitative design but it is very small for interpreting the quantitative data. In addition, in the study there were only nine participants from Japan. This is a relevant potential bias that should be discussed in the Limitations.
  2. People with severe disabilities involves a quite big and heterogeneous group, including people with physical, sensory and intellectual disabilities. The title, aims and terms used through the text refer to “people with severe disabilities”, but the participants present with physical disabilities almost exclusively. Authors should revise the terms used to appropriately name the participants and conclusions drawn from this study.
  3. Data Collection. What measures did the authors use to operationalize quantitative data? Did they use some kind of standardized scale or questionnaire about the use of community care? If they used an ad-hoc measure, it should be included as an appendix
  4. Table 1. What does “personal assistant” refer to? (not mentioned before).
  5. Table 1. Please include the quantitative data collected (i.e., used facilities, period of use and time taken to begin using them).
  6. When reporting results and drawing conclusions, the authors almost exclusively report the results from the qualitative data. I am not convinced that this study is a balanced mixed-design but rather a QUAL+quan design with a very little implication from the quantitative approach, but this is not described nor discussed in the text. It is important to note these aspects because the external validity of the findings and the generalizability of this study to other people (even from Japan and South Korean) is compromised.
  7. Considering the imbalanced, small sample used in this study, the authors should tone down their conclusions, especially taking into account the purposive sampling method, the limitations regarding participants and the obtained data.

Round 2

Reviewer 4 Report

I would like to thank the authors for their revision and reply. The revised version of the manuscript has been improved, and I think it will be an interesting addition to IJERPH.